# Achievement of the Selectivity of Cytotoxic Agents against Cancer Cells by Creation of Combined Formulation with Terpenoid Adjuvants as Prospects to Overcome Multidrug Resistance

**DOI:** 10.3390/ijms24098023

**Published:** 2023-04-28

**Authors:** Igor D. Zlotnikov, Natalia V. Dobryakova, Alexander A. Ezhov, Elena V. Kudryashova

**Affiliations:** 1Faculty of Chemistry, Lomonosov Moscow State University, Leninskie Gory 1/3, 119991 Moscow, Russia; 2Laboratory of Medical Biotechnology, Institute of Biomedical Chemistry, Pogodinskaya St. 10/8, 119121 Moscow, Russia; 3Faculty of Physics, Lomonosov Moscow State University, Leninskie Gory 1/2, 119991 Moscow, Russia

**Keywords:** cytostatic, overcoming multidrug resistance, efflux inhibitor, ion channel

## Abstract

Oncological diseases are difficult to treat even with strong drugs due to development the multidrug resistance (MDR) of cancer cells. A strategy is proposed to increase the efficiency and selectivity of cytotoxic agents against cancer cells to engage the differences in the morphology and microenvironment of tumor and healthy cells, including the pH, membrane permeability, and ion channels. Using this approach, we managed to develop enhanced formulations of cytotoxic agents with adjuvants (which are known as efflux inhibitors and as ion channel inhibitors in tumors)—with increased permeability in A549 and a protective effect on healthy HEK293T cells. The composition of the formulation is as follows: cytotoxic agents (doxorubicin (Dox), paclitaxel (Pac), cisplatin) + adjuvants (allylbenzenes and terpenoids) in the form of inclusion complexes with β–cyclodextrin. Modified cyclodextrins make it possible to obtain soluble forms of pure substances of the allylbenzene and terpenoid series and increase the solubility of cytotoxic agents. A comprehensive approach based on three methods for studying the interaction of drugs with cells is proposed: MTT test—quantitative identification of surviving cells; FTIR spectroscopy—providing information on the molecular mechanisms inaccessible to study by any other methods (including binding to DNA, surface proteins, or lipid membrane); confocal microscopy for the visualization of observed effects of Dox accumulation in cancer or healthy cells depending on the drug formulation as a direct control of the correctness of interpretation of the results obtained by the two other methods. We found that eugenol (EG) and apiol increase the intracellular concentration of cytostatic in A549 cells by 2–4 times and maintain it for a long time. However, an important aspect is the selectivity of the enhancing effect of adjuvants on tumor cells in relation to healthy ones. Therefore, the authors focused on adjuvant’s effect on the control healthy cells (HEK293T): EG and apiol demonstrate “protective” properties from cytostatic penetration by reducing intracellular concentrations by about 2–3 times. Thus, a combined formulation of cytostatic drugs has been found, showing promise in the aspects of improving the efficiency and selectivity of antitumor drugs; thereby, one of the perspective directions for overcoming MDR is suggested.

## 1. Introduction

Oncology is one of the leading causes of human mortality. Therefore, the development of effective antitumor preparations, but with minimal side effects, is an urgent task. Widely used drugs include Pac, Dox and cisplatin. Pac affects the process of cell division, stimulates the assembly of microtubules from tubulin dimers and causes the formation of multiple microtubule stars during mitosis, and this causes apoptosis [1,2,3,4,5]. It is used as a first-line medicine (ovarian, breast, etc.), often in combination with cisplatin and its derivatives. Cisplatin is used for the treatment of various solid types of cancer, exhibiting antitumor activity due to the formation of DNA damage by interacting with purine bases on DNA. However, side effects and drug resistance are two inherent problems of cisplatin, which limit its use and effectiveness [1,6,7]. Dox is used to treat lymphoblastic leukemia, sarcomas, etc. However, the drug is cardio- and nephrotoxic [8,9,10,11,12]. The mechanism of action of Dox includes [13,14]: (1) intercalation of Dox into DNA base pairs, causing DNA strands to break and inhibiting the synthesis of both DNA and RNA; (2) Dox inhibits the enzyme topoisomerase II, causing disruption of DNA unwinding for transcription and replication, DNA damage, and the induction of apoptosis.

In this paper, the focus of attention is enhancing the effectiveness of the presented drugs and reducing their side effect due to selectivity of action on cancer cells. To achieve this, it is necessary to consider (i) ways to overcome multidrug resistance (MDR) (this is primarily due to the expression of efflux proteins by tumor cells) and (ii) methods of selective drug delivery to tumors due to differences in the microenvironment and morphology of tumor and healthy cells—effects on ion channels, the use of pH-sensitive intelligent formulations that allow selectively accumulating drug molecules in tumor cells.

The main cellular and molecular mechanisms that cause resistance and MDR of cancer cells [11,15,16,17]:(1)Efflux and low accumulation of drug. Proteins involved: P-glycoprotein, TMEM205, ATP7A and ATP7B; overexpression of the ATP-binding cassette (ABC) transporter [18,19] realizing transport of various substrates across cellular membranes:MDR3 transporter [20,21] (a protein with MM of 140 kDa) is a floppase that moves phosphatidylcholine from the inner to the outer leaflet of the canalicular membrane bilayer. Lipid transporters MDR3 and MDR1 (P-gp) have several common substrates, including digoxin, Pac and vinblastine, and can cause MDR.MRP2 [20,21,22] (MDR-associated protein 2) is a unidirectional transporter that mainly transports organic anions, including drug conjugates and conjugated bilirubin. MRP2 causes resistance to various chemotherapeutic agents, such as cisplatin, methotrexate, Pac.BCRP (Breast Cancer Resistance Protein), is an efflux transporter that is generally co-expressed with MDR1, and shares many of its substrates, inhibitors and inducers. It is inhibited by some calcium channel blockers such as amlodipine, amlodipine and nifedipine.(2)Intracellular inactivation of cisplatin through binding with glutathione and metallothioneins [23];(3)Besides the efflux pump, mechanisms of resistance to Pac also include the alteration of the microtubule composition [24];(4)Insufficient sensitivity of the target, for example, topoisomerase in the case of Dox [25].

MDR is directly caused by efflux pumps and low influx. Therefore, efflux inhibitors are promising enhancers of antitumor drugs, but many of them are toxic substances (derivatives of quinoline, naphthalene and piperazine, carbonyl cyanide), so we suggest focusing on pure substances extracted from plants (such as rosemary phytochemicals [26], Thai herbs [27], allylbenzene’s derivatives and terpenoids [28,29], bicalutamide [30]). We have previously studied the process of inhibition of efflux pumps in bacterial cells of *E. coli* and *B. subtilis* by eugenol, menthol, apiol and their analogues [28,29]. These substances can likely be effective efflux inhibitors in tumor cells and agents acting on ion channels.

For EG and its analogues, anti-inflammatory, analgesic, and antimicrobial action have been shown; moreover, EG is a synergist of antibiotics [29,31,32,33,34,35,36,37,38,39,40,41,42,43,44,45,46,47,48]. In the aspect of antitumor activity about EG, it is known that this safe substance is comparable in strength to toxic cisplatin on some types of cells (e.g., U-937, HL-60, HepG2, 3LL Lewis, SNU-C5—IC_50_ is from 20 to 130 μM) [49]. In addition, the synergistic action of eugenol in combination with conventional chemotherapeutic drugs has been elucidated through some in vitro and in vivo animal model studies [50]. Moreover, allylbenzenes (e.g., apiol, EG, myristicin), mono- and sesquiterpene derivatives inhibit calcium and potassium ion channels, which ensures apoptosis of tumor cells [51]. Menthol activates the TRPM8 Ca^2+^-permeable channel in human prostate cancer cells (PC3) [52,53]. Dietary additions of (–)-menthol and other monoterpenes such as D-limonene, 1,8-cineole, α-pinene, linalool and myrcene resulted in significant inhibition of mammary carcinogenesis [53,54]. Enhanced by menthol anti-cancer activity of Pac and vincristine in HepG2 cells due to the involvement of CYP3A4 downregulation was reported [55]. Thus, adjuvants from plant extracts are promising in terms of strengthening anti-cancer drugs and overcoming multidrug resistance. Myristicin is potentially able to change the microenvironment of the tumor to a “hot” type. Apiol possibly blocks the active P-glycoprotein site, which, in turn, inhibits Dox efflux, increasing the antiproliferative response of these chemotherapeutic agents [11].

However, there remains the problem of low solubility of the allylbenzenes and terpenoids (0.01–1 mM) offered as adjuvants, and the anticancer drugs themselves. Previously, we synthesized inclusion complexes of EG and its analogues in cyclodextrins, which allowed an increase in solubility by 1–2 orders of magnitude [29]. To increase the solubility, bioavailability, and effectiveness of Pac, Dox, and cisplatin, various molecular containers are described in the literature: chitosan-based micelles and nanoparticles [8,9,12], polyethyleneimine-cyclodextrin conjugates [28], nanopolymer dextrose [56], alginate hydrogels functionalized with β-cyclodextrin [57].

In this work, β-cyclodextrins (CDs) derivatives with an external hydrophilic shell and an internal hydrophobic cavity were used to form inclusion complexes with the drugs with loading efficiently characterized by *K*_d_ 10^−3^–10^−5^ M. CDs protect the substance from destruction and inactivation, increase the half-life, and in addition, due to adsorption on cell membranes, increase the membrane permeability [29,58]. We have shown the necessity of the use of molecular containers based on CD to increase the solubility of cytotoxic agents and the possibility of using aromatic adjuvants (which are otherwise not applicable in their free form) to obtain an efficient combined antitumor formulation.

Thus, in this study, the approach to increase the effectiveness and selectivity of chemotherapy based on combined formulation of cytostatic drugs with adjuvants in the composition of inclusion complexes with β-cyclodextrins (CDs) derivatives is suggested. We studied the interaction of cytostatic formulations developed with model tumor and model healthy cells using a combination of methods to elucidate the mechanism of selectivity and proof-of-concept. Further research may be promising in aspects of solving the problem of MDR and highly selective anticancer drugs.

## 2. Results

### 2.1. Spectral Approach to the Study of the Interaction of Anticancer Drugs and EG with Cyclodextrins

Figure 1a shows the absorption spectra of Dox in the visible region in its free state and during incorporation into the MCD cavity. Since the peak is multicomponent, we carried out the approximation by Gaussians (Figure 1b) and monitored the changes in the positions of the components when the Dox is included in the MCD; a shift to the short-wave region up to 25–30 nm was observed (Figure 1c). Figure 1d shows the UV spectra of paclitaxel (Pac) in the free state and in the process of incorporation into the MCD cavity. During the transition of Pac from the ethanol environment to the hydrophobic environment of cyclodextrin, the absorption intensity increases (Figure 1e) due to the monomerization of the drug microcrystals and theoretically greater efficiency. Conversely, the optical density of cisplatin solution decreases when it is included in the CD cavity (Figure 1f).

The formation of EG inclusion complexes in the MCD was studied using FTIR microscopy (Figure 1g). FTIR spectra are recorded at 16 points and peak intensities are integrated at the MCD (C–O–C, 960–1180 cm^−1^) and EG (aromatic C–C, 1498–1534 cm^−1^) regions, which are displayed in a colored map corresponding to the quantity of components. MCD is evenly distributed on the sample with an integral intensity of ~160 units and MCD uniformly includes EG molecules (on most of the intensity map, the integral intensity is 4–5 units). Thus, the heat map of the FTIR spectra visualized the efficient incorporation of EG into the cyclodextrin. This will allow the use of substances from the group of allylbenzenes and terpenoids in soluble form as antitumor agents and cytostatic enhancers.

The absorption peak positions and maximums of absorbance can be considered as the analytical signal indicated for drug–MCD complex formation (Figure 1). Data in the Scatchard coordinates allow the calculation of the drug–MCD dissociation constants (Table 1). It is worth noting that the interaction of unmodified CD with Dox, Pac, cisplatin and vincristine is weak (*K*_d_ 10^−2^–10^−3^ M), which does not allow CD to be used as a delivery system. However, significantly better interactions with drugs were observed for modified CD (*K*_d_ 10^−4^–10^−5^ M), as presented in Table 1.

### 2.2. Anticancer Activity of Enhanced Drugs

#### 2.2.1. Effect of Cyclodextrin on Anti-A549 Drug Activity

We composed inclusion complexes of cytotoxic agents Pac, Dox, and cisplatin with MCD. Figure 2 shows the effect of the cytostatic drug inclusion in MCD on the anti-A549 activity. The most striking effect is observed for Pac: cell survival is reduced by 10–15%. For Dox, the role of cyclodextrin is noticeable at Dox concentrations of 1–10 μM. For cisplatin, the effect of the MCD is the most pronounced at high concentrations of cytostaticity, which is due to the prevention of hydrolysis of the drug (the most specific reason for cisplatin).

#### 2.2.2. Effect of Adjuvants on Anti-A549 Drug Activity

EG and analogues, as we have shown earlier, could inhibit efflux (release of drugs from the cell) and loosen the membrane of tumor cells, which presumably has an amplifying and synergistic effect on the cytotoxic activity of the substances under study (Pac, Dox, Cisplatin). Figure 3 shows the dependences of A549 cell viability on the concentration of cytostatic drugs, as well as terpenoid allylbenzene series substances. In order to estimate the synergism effect of adjuvant studied, the authors calculated the coefficients of synergism for the formation of cytostatic + adjuvant (Table 2). Cytostatic-adjuvant synergism coefficients were calculated as k = CV(cytostatic + adjuvant)/(CV(cytostatic) × CV(adjuvant)), where CV is the cell viability. According to this “synergism coefficient k”, one can distinguish interactions such as synergism (at k < 0.9), additivity (0.9 < k < 1.2), indifference (1.2 < k < 2) and antagonism (k > 2). The concept of indifference is introduced because when using two substances, competitive penetration into the cell can be observed; therefore, the value of k > 2 can be considered antagonism—as we have shown earlier on bacterial cells [29]. Synergism is found for Pac when enhanced with EG and analogues, the strongest effect is observed for safrol and EG at a concentration of 0.1–5 mM. Dox and Cisplatin show synergism with apiol at high concentration, additivity or indifference in other cases studied (Table 2).

Figure 3b visualizes the effect of the combined use of EG and Pac. Pacat a concentration of 100 nM inhibits the growth of A549 cells by 25%, while the addition of EG as an amplifier increases the effect by up to 90%. The IC50 of paclitaxel is reduced by two orders of magnitude due to EG.

Other substances such as apiol, safrol and menthol can also be used as adjuvants (Figure 3c, Table 2). Apiol at a concentration of 1.6 mM enhances the effect of Pac by 25% in the cell survival scale, and the effect is directly proportional to concentration. However, in the case of safrol and the menthol/eugenol mixture, there is an optimum: 50 μM for safrol and 25 μg/mL (150–160 μM) of EG and menthol. This effect is probably due to competition in penetration into A549 cells and a decrease in the effectiveness of the cytostatic at higher concentrations of the enhancer. Apiol has an enhancing effect on both Dox and cisplatin (Figure 3d), EG has the same effect, but with a greater concentration.

So, the strongest synergism effect is characteristic of Pac when enhanced with EG and safrol at concentration of 0.1–5 mM (Table 2). The main idea of the work is not only to strengthen antitumor drugs (Table 2) but also to partially replace toxic components with biocompatible extracts. Thus, the fact of the effectiveness of the cytostatic + adjuvant formulation has medical significance in the long term (reduction of IC50 and minimization of side effects).

### 2.3. Activity of Enhanced Drugs against Normal Cells

Figure 4 shows the results of the MTT test for HEK293T cells under the action of Pac. Pac at concentrations > 100 nM reduces cell survival to less than 30%, while the use of apiol or EG alkylbenzenes even increases the ability of healthy cells to resist cytotoxic agents. The greatest effect was achieved for a system with pre-incubated EG and apiol which penetrated the cells in advance and affected the membrane and transport proteins, reducing the flow of Pac into the cells. A similar effect was observed for cisplatin and Dox: an increase in survival from 34 to 86% and from 27 to 78% at a concentration of cytotoxic agents of 1 µM.

### 2.4. FTIR Spectroscopy as a Tool for Studying the Molecular Mechanism of Cytostatic Penetration into Cells

#### 2.4.1. Dox and EG with A549 Interaction

Figure 5 shows the FTIR spectra of A549 cells depending on the time of incubation with Dox in low and high concentrations to monitor the state of cells upon drug actions or to follow the status of the cytostatic itself, respectively, which allows you to find out which functional groups of the drug molecule are involved in interaction with the appropriate cellular components. Figure 5a shows changes in the status of A549 cells: the intensity of peaks 2850–2950 cm^−1^ increases, which corresponds to the loosening of the membrane due to penetration of Dox; the intensity of amide I increases (1630 cm^−1^) and a shoulder appears at 1640 cm^−1^, which indicates the involvement of surface and transmembrane proteins at Dox binding; a change in the peak at 1086 cm^−1^ indicates the penetration of Dox into the cells.

Figure 5b visualizes the state of Dox (at high Dox concentrations) when interacting with A549 cells. To find out the state of the A549 cells themselves under the action of EG, we used low Dox concentrations (Figure 5c). In the presence of EG, the changes similar to those described above for Figure 5a are observed; however, the position of the center of mass of the C–O–C peak of carbohydrates in the composition of cells (including DNA) is additionally shifted from 1097 to 1090 cm^−1^ (Figure 5d), which indicates much higher efficiency of Dox accumulation inside A549 and DNA intercalation.

#### 2.4.2. Dox and EG with HEK293T Interaction

Figure 6 shows the FTIR spectra of HEK293T cell suspensions depending on the incubation time with Dox and EG. The interaction of Dox-MCD (Figure 6a) with HEK293T leads to significant changes in the FTIR spectrum of cells, which, as discussed above for A549, indicates loosening of the membrane, DNA intercalation and higher availability of internal components of cells for IR radiation due to the penetration of Dox. This effect is similar to that observed for A549.

### 2.5. CLSM as a Tool for Visualizing the Penetration of Cytotoxic Agents into the Cells

Figure 7 and Appendix A show confocal images of A549 cells incubated with Dox in free form and enhanced with adjuvants. The accumulation of the drug is accompanied by an increase in cell fluorescence and the brightness of the red color in the images.

Quantitatively, the accumulation of Dox in cells is characterized by cell-associated fluorescence (Figure 8), determined by the difference in the fluorescence intensity of the cells before and after incubation with cells and by analyzing confocal images (pixel intensity integral). For cancer cells, accumulation over a time of 15 min is given in the manuscript (Figure 7), and for the rest of the time intervals—in the Appendix A.

For HEK293T, the observed differences are minor even after 45 min incubation and quite bright after 120 min because the accumulation of the drug is slow, and after 15 min, the drug practically does not penetrate (Figure 8b).

## 3. Discussion

### 3.1. Interaction of Anticancer Drugs and EG with Cyclodextrins

Spectral methods provide information about the changes in microenvironment of functional groups upon intermolecular interactions. A wide range of methods for determining parameters of drug–cyclodextrin interactions are described in the literature, including FTIR [39,59], NMR spectroscopy [59,60,61,62,63,64], fluorescence [65,66,67,68], and electrochemical methods [69]. Molecular absorption spectroscopy using mathematical functionality is a worthy method of tracking the inclusion of various drugs in the hydrophobic cavity of cyclodextrin. It is known that aromatic molecules form non-covalent complexes with CD of the order of 10^−2^–10^−6^ M, which makes it possible to create, on the one hand, a simple, but at the same time effective shell for the drug. The inclusion of antitumor drugs in CD is promising in terms of enhancing penetration into cancer cells and the possibility of using adjuvants from plant extracts.

The dissociation constants obtained for Dox and Pac complexes with MCD (Table 1) indicate the prospects of cyclodextrins as protective molecular containers for the studied antitumor drugs. Dox interacts with MCD in a ratio of 1:4, which indicates to the stabilization of the complex due to MCD- Dox-MCD interaction possibly in the form oligomers of rings, stars or other supramolecular architecture [70,71]. Pac interacts with an equimolar amount of MCD, mainly due to the inclusion of only benzene rings in the cyclodextrin cavity, while the rest of the drug (1,7-dihydroxy-9-oxo-5,20-epoxytax-11-en-2-yl) interacts with the outer shell of the MCD (Me- and OH-groups). These data are in good agreement with the calculated parameters described in the paper [72]: methylation of hydroxyl groups makes complexation possible only through a large rim of CD torus; the most favored phenyl rings for the inclusion is the benzoate moiety in the 2 position of the Pac and phenyl in 3′. For cisplatin, the protective container is extremely important, since hydrolysis proceeds quite quickly: *K* = [Pt(NH_3_)_2_(H_2_O)Cl]^+^ × [Cl^−^]/[Pt(NH_3_)_2_Cl_2_] = 0.0037 M [73]. It is slowed down only at a concentration of chloride ions of the order of 1 M, which is unattainable upon application, so loading cisplatin into MCD could provide protection from hydrolysis and higher efficacy of the drug.

### 3.2. Anticancer Activity of Enhanced Drugs

In this paper, we focused on studying the effect of the CD molecular container and formation of double complexes of cytotoxic agents and the adjuvants of a series of allylbenzenes and terpenoids (eugenol, apiol, menthol, and safrol) on the antitumor activity of Pac, Dox, and cisplatin. A549 (adenocarcinomic human alveolar basal epithelial cells) cells were selected as a target for anticancer drug.

The use of cytostatic drugs in free form is limited due to low solubility and possible degradation of the drug. Therefore, drug carriers are used to increase the bioavailability and effectiveness. CDs are one of the simplest molecular containers (what is important for implementation in medical practice and production), at the same time increasing the effectiveness of drugs. It is known from the literature that for several drugs the mechanism of action of cyclodextrin consists in increased adsorption on the cell wall and increased penetration of the drug inside [74]. This is implemented for the systems we are considering.

In the introduction, the authors discussed in detail adjuvants and their potential utility as enhancers of anticancer drugs. Here are some experimental confirmations on the cytostatic+adjuvant systems studied by us in molecular containers. The most powerful cytostatic agent against lung carcinoma is Pac (IC50 = 630 nM). This is followed by Dox with IC50 = 450 nM, but Dox reduces the survival of A549 cells by no more than 70%. Cisplatin acts only in high concentrations (IC50 = 50 µM), which causes its toxicity to the body. A curious fact turned out to be that EG and analogues (in mM concentrations) have pronounced antitumor activity comparable in strength to reference cytotoxic agents (there were indications to this effects in the literature too).

Pac at a concentration of 100 nM inhibits the growth of A549 cells by 25%, while the addition of EG as an amplifier increases the effect by up to 90%. The IC50 of Pac is reduced by two orders of magnitude due to EG.

Apiol has an enhancing effect on both Dox and cisplatin (Figure 3d): a decrease in the survival rate of A549 cells from 84 to 40% and from 80 to 7%, respectively. EG has the same effect, but with a greater concentration.

Thus, it is shown that cytostitic drugs in the form of complexes of inclusion in the MCD and enhanced with adjuvants are much more effective than simple formulations and promising for use in medical practice.

### 3.3. Activity of Enhanced Drugs against Normal Cells

An important aspect is not only the strengthening of the antitumor activity of drugs but also selectivity; ensuring that the developed formulation acts to a lesser extent against healthy cells is critical. Ideally, the combined drug should attack cancer cells more strongly and not damage healthy cells. Such “smart” recognition of one’s own and another’s is possible due to the morphological differences between cancer and healthy cells, which deftly use delivery systems (cyclodextrins) in combination with additional enhancer substances (EG and analogues).

Cancerous cells differ from normal cells in terms of cell growth, morphology, cell–cell interaction, organization of the cytoskeleton, and interactions with the extracellular matrix. Normal cells have a single-length brush of ~2.4 µm with a grafting density of ~300 ‘molecules’ per µm^2^, whereas cancerous cells have a brush with two characteristic lengths of 0.45 and 2.6 µm with grafting densities of ~640 and 180 ‘molecules’ per µm^2^, respectively [75], which creates heterogeneity of the surface of cancer cells and can cause the formation of defects under the action of adjuvants. The cell stiffness of metastatic cancer cells is more than 70% softer than the benign cells that line the body cavity; meanwhile, different cancer types were found to display a common stiffness [76]. Therefore, it is expected that the effect of adjuvants on the membrane permeability will have a greater effect on tumor cells, while it will have a protective effect on healthy ones. Ion channels contribute to a variety of processes such as maintenance of cellular osmolarity and membrane potential, motility, invasion, signal transduction, transcriptional activity and cell cycle progression, leading to tumour proliferation and metastasis [77,78]—therefore, they are promising targets for the action of selective inhibitors, which may be EG and its analogues [51,52,53,54,55].

The protective effect of combining drugs may presumably be due to the action of allylbenzenes on reduced membrane permeability, closing membrane defects and making it more elastic (the anti-inflammatory properties of EG and apiols, which are based on this, are known). How did it happen that in healthy cells the EG protects the membrane? Previously, on model systems, liposomes, we observed that IG-CD really stabilizes the lipid membrane by closing defects. On the contrary, tumor cells have large pores (link) and they skip EG-SD and inhibit efflux there) [79,80].

### 3.4. FTIR Spectroscopy as a Tool for Studying the Molecular Mechanism of Cytostatic Penetration into Cells

FTIR spectroscopy can be effectively used to monitor the molecular details of the interactions of drug with cells—a technique recently developed by the authors [81], in which the authors clearly showed that the method is sensitive: if drugs affect cells, then the IR spectra change greatly, and, on the contrary, minimal changes are observed if the drug does not work. In the cell, it is possible to distinguish the main structural units that contribute to the absorption of IR radiation (Figure 5a): cell membrane lipids (2800–3000 cm^−1^); proteins, especially transmembrane (1500–1700 cm^−1^); DNA phosphate groups (1240 cm^−1^); and carbohydrates, including lipopolysaccharides (900–1100 cm^−1^) [81]. The FTIR spectra of lipids and phospholipids has the following characteristic peaks of functional groups: two bands of symmetric and asymmetric oscillations of hydrocarbon bonds, oscillations of the carbonyl group C=O, and oscillations of the phosphate (Figure 5a). The position of the bands and their shape are sensitive to binding of the bilayer with ligands or drug molecules, hydrogen bond formation, aggregation and oxidation, etc. [82].

Without eugenol, the changes in the FTIR spectra of A549 (Figure 5) are insignificant: from the red to the blue spectrum. After the addition of EG, the FTIR spectrum begins to change dramatically (from yellow to brown and gray): a decrease in peak intensity by 1630 cm^−1^ indicates the penetration of Dox into cells; an increase in peak intensity by 1505–1515 cm^−1^ indicates the interaction of eugenol with the cell membrane and transport proteins (which may correspond to interaction with ion channels or inhibition of efflux proteins); an increase in the intensity of peaks in the region of 1240–1280 cm^−1^ indicates intercalation of DNA with Dox inside A549.

However, it is interesting that the effect of EG on healthy cells HEK293T (Figure 6b) practically does not lead to a change in the structure of cells and affects their growth, since the IR spectrum change is extremely weak. Moreover, pre-incubated EG, which has penetrated HEK293T cells (Figure 6c), reduces the effectiveness of Dox penetration. Thus, EG and its analogues provide the selectivity of cytostatic against cancer cells, additionally protecting healthy cells.

In analytical terms, we can also draw conclusions: an integrated approach based on a combination of methods works, and direct correlations with MTT test data are observed; in addition, FTIR spectra really reflect changes in cells (when there are changes, we clearly see them, and when they are not, then they are not visible in the spectra).

### 3.5. CLSM as a Tool for Visualizing the Penetration of Cytotoxic Agents into the Cells

To clarify the correlation of the data obtained by the methods of FTIR spectroscopy and MTT test allowing to directly determine cell survival, we used confocal laser scanning microscopy (CLSM), which allows us to study the effect of molecular containers and adjuvants on the effectiveness of cytotoxic agents against cancer cells with our own eyes. The aim of the CLSM experiment is to compare the accumulation of simple Dox with the combined form of Dox+adjuvant. Moreover, the effect of the adjuvant (enhancer—eugenol and apiol) was studied in two modes: (1) major membrane permeability enhancer and minor efflux inhibitor, when EG or apiol were simultaneously added with Dox; (2) minor membrane permeability enhancer and major efflux inhibitor, when EG or apiol were preincubated with A549.

Free Dox accumulates weakly in A549 cells (Figure 7a and Appendix A). Dox is released from the cell by pumping proteins in a process of efflux. When an adjuvant is included in the system simultaneously with Dox, an increase in the degree of Dox accumulation in cells is observed (Figure 7b,c and Appendix A) due to the occurrence of defects in the cell membrane and inhibition of pump proteins—the adjuvant is active, but not in full force. The effect of adjuvants is enhanced when they are pre-incubated with cells and thus accumulated inside with inhibited efflux pumps and loosened the cell membrane.

Quantitatively, the accumulation of Dox in cells is characterized by cell-associated fluorescence (Figure 8). At a time interval of 15 min, adjuvants already demonstrate their potential: due to EG or apiol, Dox penetrates 2 times more efficiently, while in a system with pre-incubated eugenol, the accumulation of Dox is 3.5 times higher. Accumulation of Dox as a function of time: free Dox accumulates extremely slowly and reaches a plateau after 1 h, and after several hours, it may decrease due to efflux. In systems with alkylbenzenes, a higher concentration is maintained 2–3 times in relation to free Dox, practically independent of time.

Cardinally different situation is observed for healthy HEK293T cells. EG and apiol reduce the accumulation of Dox in healthy cells by 20–30%, and pre-incubated adjuvants are much more effective.

Thus, a correlation is shown between the data obtained by MTT test methods, FTIR spectroscopy with what is observed in a confocal microscope and quantified by fluorescent methods.

## 4. Materials and Methods

### 4.1. Reagents

Doxorubicin (Dox) hydrochloride and methyl-β-cyclodextrin (MCD) were obtained from Sigma Aldrich (St. Louis, MI, USA). Cisplatin was obtained from Teva Pharmaceutical Industries (Tel Aviv, Israel). Eugenol at the highest commercial quality was purchased from Acros Organics (Flanders, Belgium). The preparation of apiol and plant extracts was carried out in the same way as described earlier [29]. Menthol and linalool were purchased from Rotichrom GC (Carl Roth GmbH + Co. Karlsruhe, Germany). Other chemicals including organic solvents, salts and acids were produced by Reakhim (Moscow, Russia).

### 4.2. MCD Inclusion Complexes Synthesis

Cytostatic and adjuvant inclusion complexes in MCD synthesis were utilized as described in an earlier paper [29].

### 4.3. Cell Cultivation and Toxicity Assay

Adenocarcinomic human alveolar basal epithelial cell A549 cell lines (Manassas, VA, USA) were grown in RPMI-1640 medium (Gibco, Thermo Fisher Scientific Inc., Waltham, MA, USA) supplemented with 5% fetal bovine serum (Capricorn Scientific, Ebsdorfergrund, Germany) and 1% sodium pyruvate (Paneco, Moscow, Russia) at 5% CO_2_/95% air in a humidified atmosphere at 37 °C. Cell lines were tested for mycoplasma contamination before the experiment using the Mycoplasma Detection Kit PlasmoTest™ (InvivoGen, San Diego, CA, USA).

Linear cells of the embryonic kidney human epithelium (HEK293T) are cultured in DMEM medium with 4.5 g D-glucose (Life Technologies, Carlsbad, CA, USA) supplemented with 10% fetal bovine serum (FBS) (Gibco, Grand Island, NY, USA) and 100 units/mL of penicillin and streptomycin. Cell passage occurred upon reaching a 70–90% confluent monolayer. The following conditions are maintained in the incubator: temperature 37 °C, 5% CO_2_ in air at constant humidity. Removal of cells from culture plastic is carried out using 0.05% trypsin/EDTA solution (Hyclone, Logan, UT, USA).

The cell line was obtained from Lomonosov Moscow State University Depository of Live Systems Collection and Laboratory of Medical Biotechnology, Institute of Biomedical Chemistry (Moscow, Russia).

To test acute toxicity, cells were cultivated for 72 h in a V-bottom 96-well plate (TPP, Trasadingen, Switzerland) in the presence of Pac, Dox, cisplatin and its combined formulations, and the cell viability was tested by measuring the conversion of the tetrazolium salt, 3-(4,5-dimethyl-thiazol-2-yl)-2,5-diphenyltetrazolium bromide (Serva, Heidelberg, Germany), to formazan (MTT test) [83,84].

Cytostatic-adjuvant synergism coefficients were calculated as CV(cytostatic + adjuvant)/(CV(cytostatic) × CV(adjuvant)), where CV is the cell viability. There are interactions such as synergism (k < 0.9), additivity (0.9 < k < 1.2) and indifference (1.2 < k < 2), antagonism (k > 2) [29].

### 4.4. UV–Visible Spectroscopy to Determine the Parameters of the Interaction of Cytotoxic Agents with MCD

UV–visible spectra of cytotoxic agents alone and in complexes with MCD in different molar ratios were recorded on the AmerSham Biosciences UltraSpec 2100 pro device (USA); 0.5 mM HCl. The background spectrum was subtracted as a blank.

Consider the equilibrium: cytostatic(aq) + n MCD(aq) ↔ cytostatic·nMCD(aq), where *K*_d_ = [MCD(aq)]^n^ × [cytostatic(aq)]/[ cytostatic·nMCD(aq)].

Calculation of the dissociation constants cytostatic–MCD based on the dependences of the position or intensity of the maxima in the absorption spectra of cytotoxic agents on the concentration of MCD and subsequent linearization in Scatchard coordinates: [cytostatic·nMCD(aq)]/[MCD(aq)] versus [cytostatic·nMCD(aq)]—the slope equals to −1/*K*_d_. The state of a free cytostatic agent was taken as the initial value, and the state at a 1000-fold excess of MCD was taken as the final value.

Entrapment efficiency of cytostatic: EE (%) = 100 × (cytostatic amount in form of inclusion complex with MCD)/(total cytostatic amount) = 100 × [cytostatic·nMCD(aq)]/([cytostatic(aq)] + [cytostatic·nMCD(aq)]) = 100 × [MCD]n/(*K*_d_ + [MCD]n).

### 4.5. FTIR Spectroscopy

#### 4.5.1. The Study of Dox and Adjuvant Actions on A549 and HEK293T Cells

ATR-FTIR spectra of the cell sample suspensions were recorded using a Bruker Tensor 27 spectrometer equipped with a liquid-nitrogen-cooled MCT (mercury cadmium telluride) detector. Samples were placed in a thermostatic cell BioATR-II with ZnSe ATR element (Bruker, Mannheim, Germany). FTIR spectra were acquired from 850 to 4000 cm^−1^ with a 1 cm^−1^ spectral resolution. For each spectrum, 50 scans were accumulated and averaged. Spectral data were processed using the Bruker software system Opus 8.2.28 (Bruker, Germany).

A549 or HEK293T cell suspensions (1.5 × 10^6^ cells/mL) were washed twice with sterile PBS (pH = 7.4) from the culture medium by centrifuging (Eppendorf centrifuge 5415C, 10 min, 3000× *g*). The cells are precipitated by centrifugation and separated from the supernatant, washed twice and resuspended in PBS (5 × 10^6^ cells/mL) to register FTIR spectra.

Cell suspensions were incubated with Dox-containing samples and FTIR spectra were registered at 37 °C online or after 0.5–1–2–3 h incubation. To quantify the absorbed Dox, the cells were precipitated by centrifugation and separated from the supernatant, washed twice and resuspended in 50 µL PBS to register FTIR spectra. The supernatant was separated to determine the amounts of unabsorbed substances.

#### 4.5.2. FTIR Microscopy for Study EG Inclusion in MCD

ATR-FTIR spectra of EG-MCD placed on KBr glass were recorded using a Bruker Lumos II IR microscope in the region from 700 to 4000 cm^−1^ with 1 cm^−1^ spectral resolution with scanning in the area on average 1 × 1 μm.

### 4.6. Confocal Laser Scanning Microscopy of Dox and Adjuvant Actions on A549 and HEK293T Cells

A549 and HEK293T cells were precipitated as described above followed by 15-, 45- or 120-min incubation with Dox 5 μg/mL: (a) free, (b) with EG, (c) with apiol, (d) with preincubated EG for 30 min, or (e) with preincubated apiol for 30 min. The cells were centrifuged twice with PBS washing (10 min, 4000× *g*). The cell centrifuge was suspended in 200 µL of PBS, followed by the addition of 100 µL of a glycerin.

Fluorescence images were obtained using an Olympus FluoView FV1000 confocal laser scanning microscope (CLSM) equipped with both a spectral version scan unit with emission detectors and a transmitted light detector. CLSM is based on the motorized inverted microscope Olympus IX81. Emission fluorescence spectra of Dox were obtained by CLSM. An excitation wavelength of 488 nm (multiline Argon laser) and dry objective lens Olympus UPLSAPO 40× NA 0.90 were used for the measurements. The laser power, sampling speed, and averaging were the same for all image acquisitions. The scan area was 80 × 80 µm^2^. Dox fluorescence was collected using the emission windows set at 540–620 nm, at 488 nm excitation. The signals were adjusted to the linear range of the detectors. Olympus FV10 ASW 1.7 software was used for the acquisition of the images.

### 4.7. Dox’ Cell Uptake Determination

Quantitative analysis of the Dox content in A549 and HEK293T was performed using fluorescence spectroscopy. λ_exci_ (Dox) = 488 nm, λ_emi_ (Dox) = 595 nm. Registration of fluorescence spectra was carried out using a Varian Cary Eclipse spectrofluorometer (Agilent Technologies, Santa Clara, CA, USA) at 22 °C. The concentration of Dox inside the cells was calculated from the material balance considering the unabsorbed fluorophore’s concentration determined by the fluorescence intensity. Intracellular concentrations of fluorophores were determined after destruction of cells by 10 min of incubation with 1% Triton X-100 solutions.

### 4.8. Statistical Analysis

The statistical residual difference between the curves for a free cytostatic and cytostatic-MCD or cytostatic + adjuvant was analyzed using a one-sample *t*-test or a point-by-point comparison was applied based on a series of repeated experiments (from three to five) with the verification of the null hypothesis and gave the largest *p*-value for the curve. Statistical analysis of the obtained data was carried out using Origin 2022 software (OriginLab Corporation, Northampton, MA, USA).

## 5. Conclusions

The problem of oncological diseases is acutely marked in the world: high mortality from cancer and the likelihood of relapses, as well as multiple drug resistance, which causes the ineffectiveness of seemingly strong antitumor drugs. In this paper, combined formulas are proposed: cytostatic + enhancer (EG, apiol—efflux inhibitors—affect the ion channels of cancer cells and increase the permeability of the cell membrane) as part of molecular containers to protect the drugs from destruction and increase the bioavailability and allows for synergism with adjuvants. EG and analogues are poorly soluble in water, while their antitumor and tissue-repairing properties are known; thus, this article, through the introduction of soluble forms of adjuvants, opens up the potential of natural extracts. We have shown that EG and apiol in mM concentrations have strong antitumor effects. The same substances in micromolar concentrations are synergists of cytotoxic agents. At the same time, EG and apiol have a protective effect on healthy cells in the HEK293T model. Based on the literature data, the probable explanation lies in the larger size of tumor cells, their greater permeability, and the selective effect of EG and apiol on ion channels and transport proteins of tumor cells.

The work uses an integrated approach—a combination of methods, from different angles and aspects revealing the secrets of drug interaction with cells. The MTT test provides quantitative data on survival, FTIR spectroscopy provides molecular details, CLSM enables cell imaging, and fluorescence spectroscopy allows the quantification of absorbed cytotoxic agents. According to the FTIR spectra of A549 cells, loosening of the membrane due to the penetration of doxorubicin, the participation of surface and transmembrane proteins in the binding of doxorubicin, and the penetration of cytostatic into cells was observed. Without EG, changes in the FTIR spectrum of A549 cells are insignificant, and after the addition of EG, the FTIR spectrum begins to change sharply due to the interaction of EG with the cell membrane and transport proteins (inhibition of efflux). It has been shown that changes in the FTIR spectra occur only when the state of the cell components actually changes. According to FTIR spectroscopy and confocal microscopy, EG reduces the accumulation of cytostatic in healthy HEK293T cells, while increasing the accumulation in cancer cells A549. The effect of EG and Apiol on efflux pumps was shown in the CLSM experiment when comparing the simultaneously added cytostatic and adjuvant to cells and the adjuvant pre-incubated with cells: it takes up to 30–60 min for the penetration of EG into cancer cells and effective inhibition of pump proteins.

Thus, the effectiveness and selectivity of doxorubicin, cisplatin and Pac have been significantly increased, which is promising in terms of prospects in overcoming multidrug resistance and creating low-toxicity drugs.

## Figures and Tables

**Figure 1 ijms-24-08023-f001:**
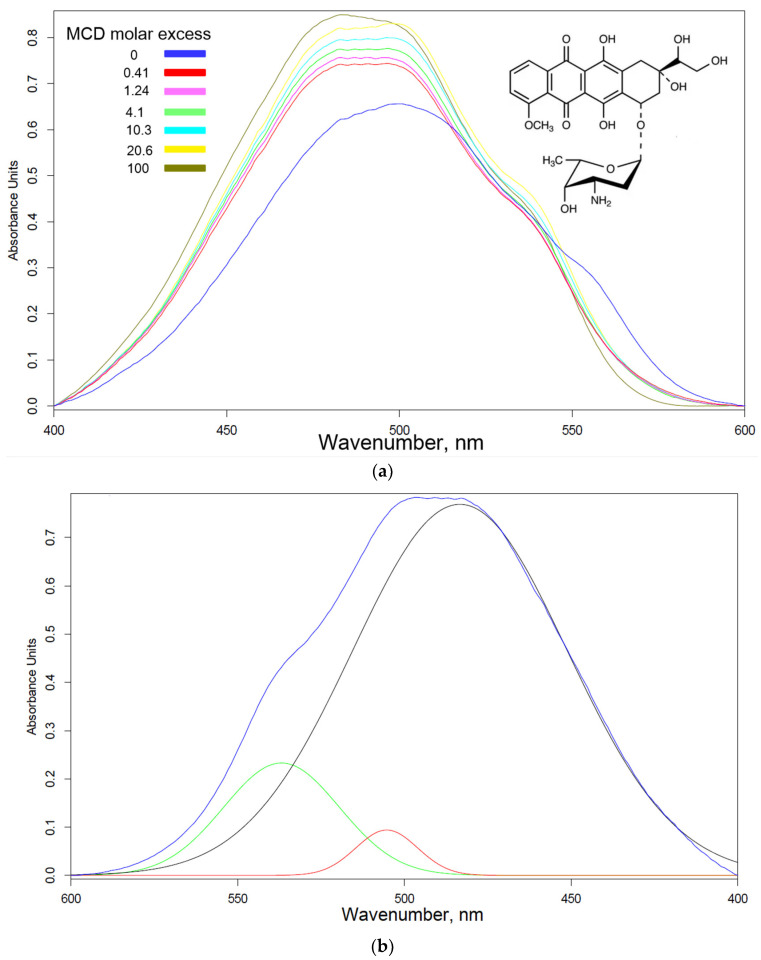
(**a**) Absorption spectra in the visible region of Doxorubicin (Dox) in free form and in complexes with MCD. (**b**) Self-deconvolution of Dox-MCD (2.5:1 molar ratio) absorption spectra with Gaussian components. (**c**) Changes in the components of (**b**) when Dox is included in the MCD cavity and linearization of data in Scatchard coordinates. (**d**) UV spectra of Paclitaxel (Pac) in free form and in complexes with MCD. (**e**) Dependence of A_230_ in (**d**) when Pac is included in the MCD cavity and linearization of data in Scatchard coordinates. Theta is the degree of inclusion of the drug in the MCD cavity. Pac was dissolved in 50% EtOH. For (**a**–**e**): 0.5 mM HCl. (**f**) UV spectra (baseline correction was applied for peaks separation) of cisplatin in free form and in complexes with MCD; in the insert, the linearization of data in Scatchard coordinates are presented. For only cisplatin: 1 M NaCl. (**g**) Solid-phase FTIR spectra of EG, MCD and EG-MCD inclusion complex. For EG-MCD, the maps of integral intensity MCD (960–1180 cm^−1^) and EG region (1498–1534 cm^−1^) in 16 points are presented. T = 22 °C.

**Figure 2 ijms-24-08023-f002:**
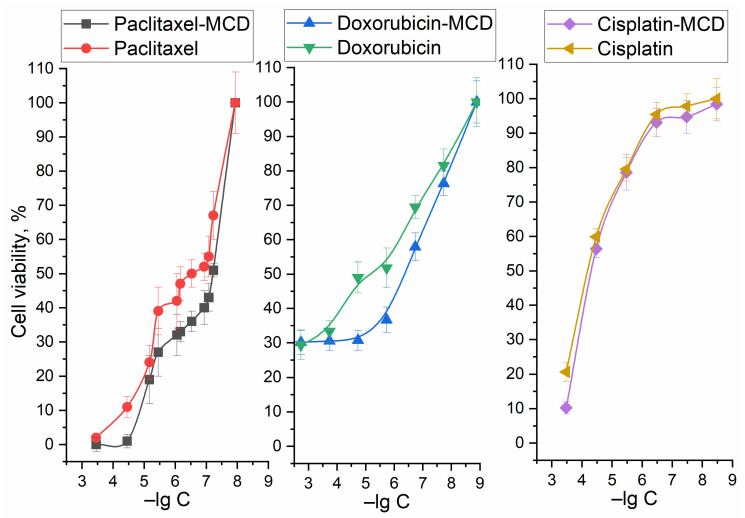
Dependences of A549 cell survival on the concentration of Pac, Doc and cisplatin in free form and in the form of inclusion complexes in MCD. RPMI-1640 medium supplemented with 5% fetal bovine serum and 1% sodium pyruvate at 5% CO_2_/95% air in a humidified atmosphere at 37 °C. The statistical residual difference between the curves for a free cytostatic and cytostatic-MCD was analyzed using one-sample *t*-test: *p*-values 9.8 × 10^−5^, 0.032 and 0.048 < 0.05 for Pac, Dox and cisplatin, respectively. –lg C (negative logarithm of the concentration of the test substance). Significance level *p* = 0.05.

**Figure 3 ijms-24-08023-f003:**
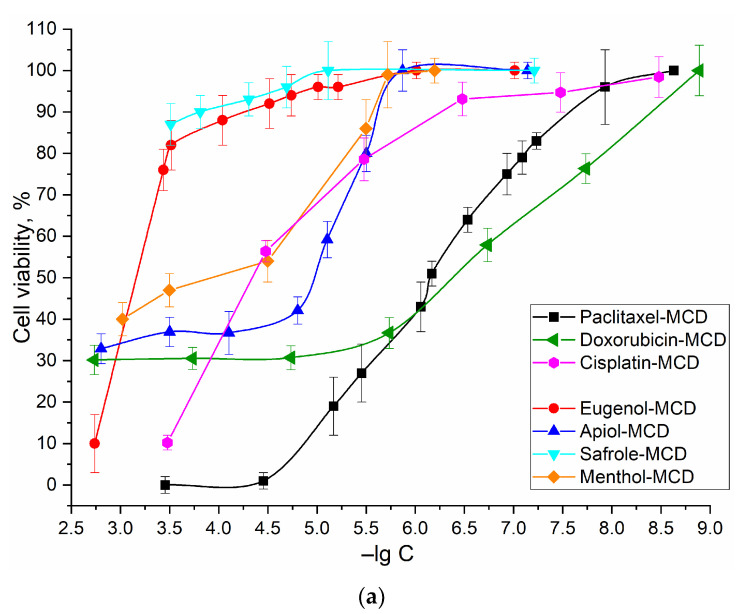
Dependences of A549 cells survival on the logarithm of concentration of: (**a**) Pac, Doc, cisplatin, EG, menthol, safrole and apiol; (**b**) EG and Pac in mixture; (**c**) adjuvants with Pac mixtures; (**d**) adjuvants with cisplatin and Dox mixtures. Color scales for substances marked with the corresponding colors. The conditions are the same as indicated in the caption to Figure 2. All compounds are used in the form of complexes with MCD. For (**c**,**d**) graphs: the statistical residual difference between the data for a free cytostatic and cytostatic + adjuvant was analyzed using one-sample *t*-test. –lg C (negative logarithm of the concentration of the test substance). The blue plane on the 3D graph (**b**) indicates the cross section at Pac concentration of 100 nM—how cell survival changes with varying adjuvant concentrations. It shows a significant increase in cytotoxic effect and synergistic effect. Significance level *p* = 0.05.

**Figure 4 ijms-24-08023-f004:**
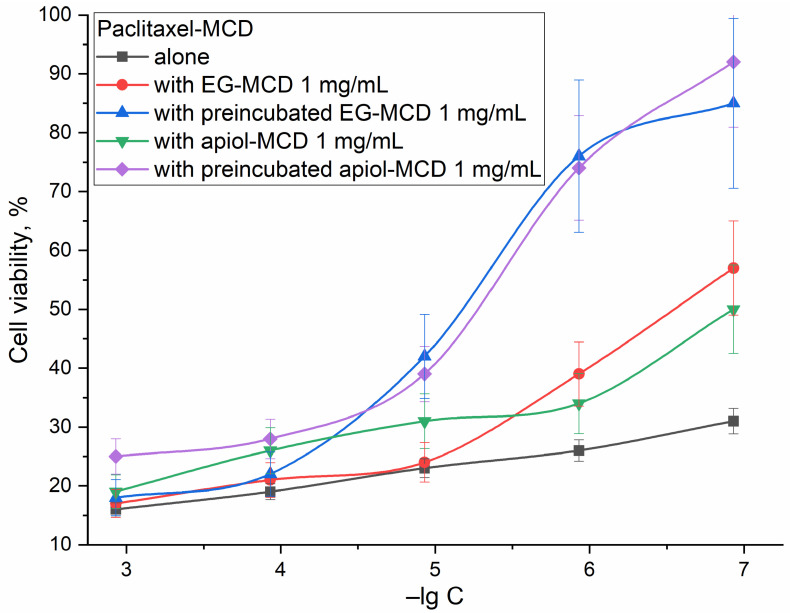
Dependences of HEK293T cell survival on the logarithm of concentration of Pac alone and in mixture with EG 1 mg/mL or apiol 1 mg/mL simultaneously added and in the mode of pre-incubation of adjuvants. The conditions are the same as indicated in the caption to Figure 2. Preincubation of EG and apiol for 1 h. All compounds are used in the form of complexes with MCD; lg C (logarithm of the concentration of the test substance).

**Figure 5 ijms-24-08023-f005:**
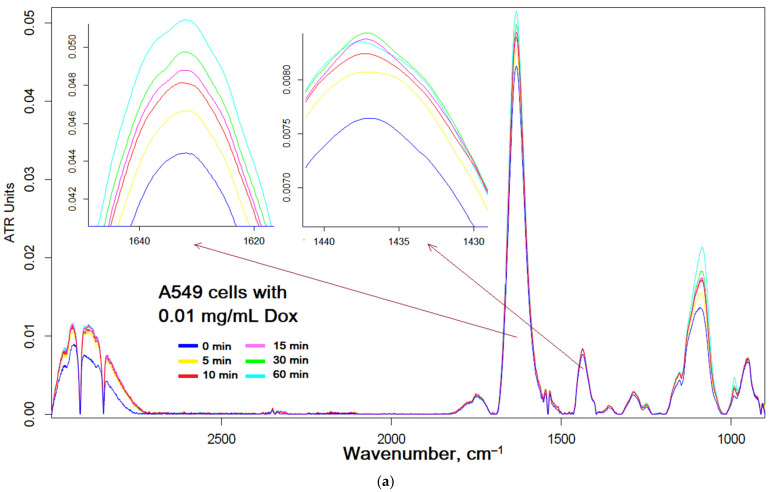
FTIR spectra of A549 cells’ suspensions (10^6^ cells/mL) during incubation with: (**a**) Dox 0.01 mg/mL, (**b**) Dox 1 mg/mL and EG 0.1 mg/mL added after 20 min of incubation, (**c**) Dox 0.01 mg/mL and EG 0.1 mg/mL added simultaneously. (**d**) Dependence of the center mass position of the peak of C–O–C oscillations on time of incubation for graphs (**a**,**c**). All compounds are used in the form of complexes with MCD. T = 37 °C. The statistical residual difference between the curves for (**d**) is significant; *p*-value is 0.0021.

**Figure 6 ijms-24-08023-f006:**
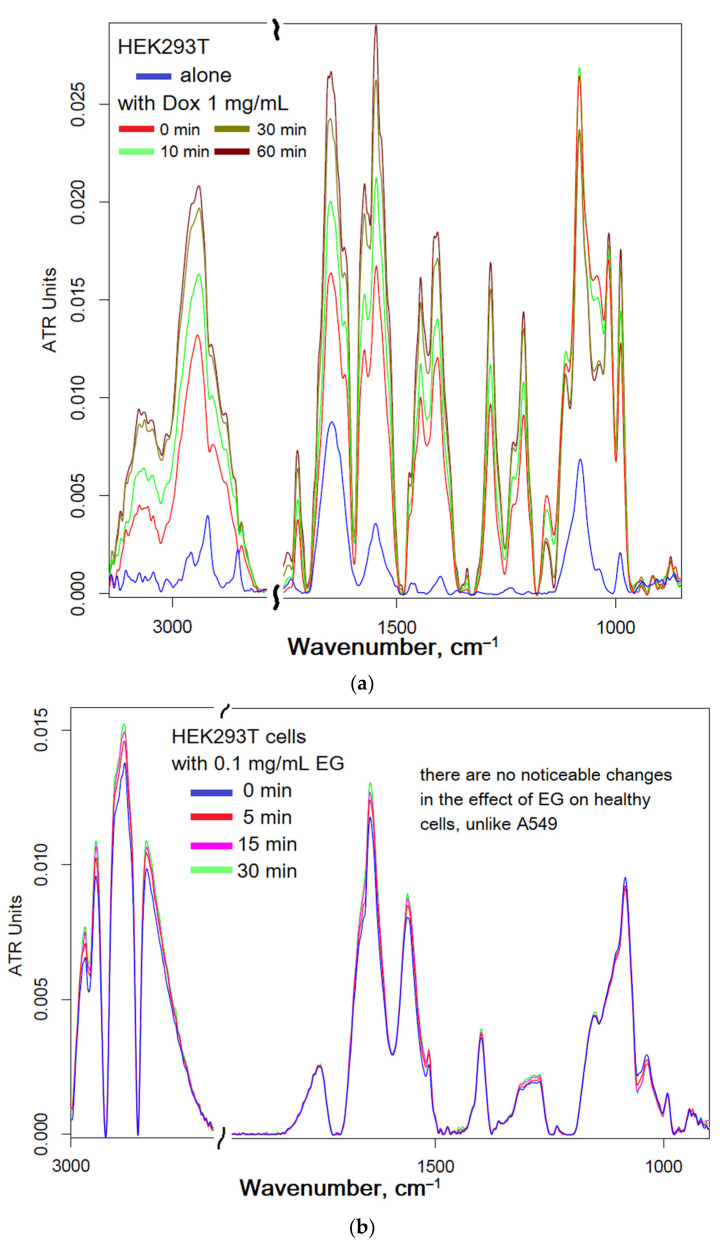
FTIR spectra of HEK293T cells’ suspensions (10^6^ cells/mL) during incubation with: (**a**) Dox 1 mg/mL, (**b**) EG 0.1 mg/mL, (**c**) Dox 1 mg/mL added after 30 min incubation of cells with EG 0.1 mg/mL. All compounds are used in the form of complexes with MCD. T = 37 °C.

**Figure 7 ijms-24-08023-f007:**
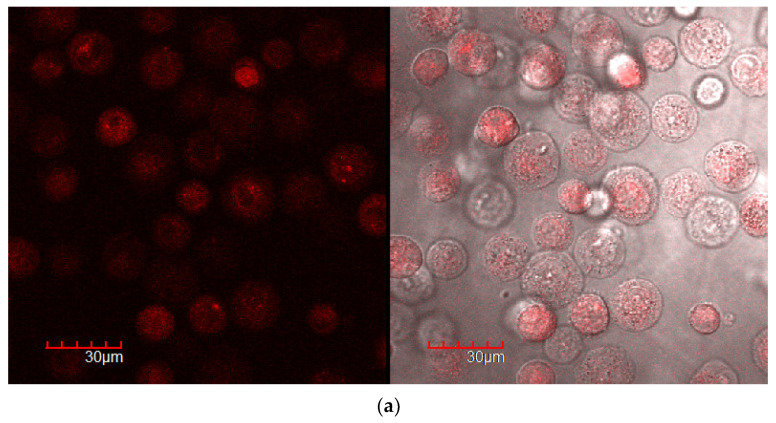
Confocal laser scanning images of A549 after 15 min incubation with Dox 5 μg/mL: (**a**) free, (**b**) with EG, (**c**) with apiol, (**d**) with preincubated EG for 30 min, (**e**) with preincubated apiol for 30 min. The scale segment is 30 µm (division value is 6 µm); 2 channels are shown: red, Dox; and overlay transmission light mode with Dox channel. λ_em_ = 488 nm (multiline Argon laser). CLSM images for 45 min and 2 h are given in Appendix A.

**Figure 8 ijms-24-08023-f008:**
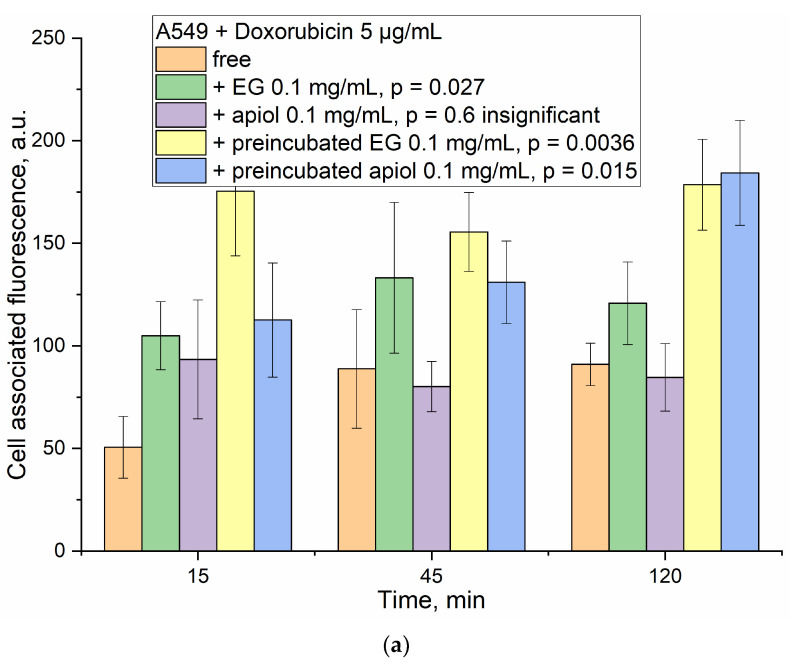
(**a**) A549-associated and (**b**) HEK293T-associated fluorescence (conventional units) depending on the incubation time and the composition of the Dox-containing formulation. PBS (0.01 M, pH 7.4). T = 37 °C. The statistical residual difference between data for control Dox and Dox + adjuvant is significant for 3 formulations, *p* < 0.05.

**Table 1 ijms-24-08023-t001:** Dissociation constant of drug–MCD complexes determined by UV–vis method; 0.5 mM HCl. T = 22 °C.

Compound X	*K*_d_ (X-MCD), μM	Entrapment Efficiency, % *
Dox	160 ± 15	87 ± 6
Pac	180 ± 20	85 ± 7
Cisplatin	130 ± 10	88 ± 6
EG	3600 ± 200	22 ± 2

* at C(MCD) = 1 mM.

**Table 2 ijms-24-08023-t002:** Coefficients of cytostatic-adjuvant synergism when acting on cancer cells A549.

	Pac		Dox	Cisplatin
Apiol	0.78, synergism	Apiol	0.3–1.9 (0.3 at C > 1 mM), synergism and indifference	0.3–2.3 (0.3 at C > 1 mM), synergism and indifference
Safrole	0.5, synergism			
EG + Menthol	0.81, synergism	EG	1.1, additivity	1.0, additivity
EG	0.9 (0.6 at C > 0.1 mM), synergism and additivity			

## Data Availability

The data presented in this study are available in the main text and Appendix A.

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
