# Peer review of "Achievement of the Selectivity of Cytotoxic Agents against Cancer Cells by Creation of Combined Formulation with Terpenoid Adjuvants as Prospects to Overcome Multidrug Resistance"

_ijms, 2023, doi:10.3390/ijms24098023_

Round 1
Reviewer 1 Report
Dear Authors,
I have found some things which you should change:
1. In introduction:
a) What is mechanism of action of doxorubicin?
b) Write more informations about ATP-binding casette transporters - MDR3 and MRP2
c) What about BCRP?
d) if you use abbrevations, be consistent in all paper:
DOX - doxorubicin
CIS - cisplatin
PAC - paclitaxel
use these abbrevations though the whole article
2. Results and discussions - should be separated ,
separated chapter for results and separated chapter for discussion
discussion - more literature references
3. Figure 3 - the whole figure should be on one page and c) is not readable - the legend should be putted in another place, not on the chart
4. I saw some mistakes in English language - the language must be verified by native
Reviewer 2 Report
This article tried to propose a novel combinatorial formulation to overcome multidrug resistance in cancer. The idea is good, however, the experimental platform, data analyses, and the interpretation of the results are not that correct. Following suggestions are provided for a better research article of the present study.
1. The whole article has numerous errors. For example, typing error, line 117, “anticancer drugs ang EG”, what dose “ang” mean? Line 241, “enhances the effect of paclitasel by”, do you mean “paclitaxel”? Line 247-248, “a decrease in the survival rate of A549 cells from 40 to 84%”, do you mean form 40 to 84 is a decreasing effect? A careful check in the typing, grammar, and the data report of the article is suggested. Also, I found the title of the manuscript and the supplementary files were different, so what title are you plan to use?
2. Another problem is the statistical analyses. The statistical significance can only be claimed when the statistical analyses are performed. I didn’t see any description of statistical analyses in the article, not in the methods or in the results and discussion. The author only stated the increasing or decreasing percentages of the results, but what about whether the results are significant? Without statistical analyses, the conclusions cannot be made.
3. The author mentioned “synergistic” effects in the article several times, however, I didn’t see any analytical experiments performed on the synergistic effects. How do you statistically know the results are synergistic, not additional? You need to perform combinatorial analyses formally, not interpreting by yourself. Many combinatorial analyses are available, you are suggested to survey related papers to learn how to formally do the combinatorial analyses.
4. Finally, the most important concern, the author claimed this novel formulation is promising for overcome “multidrug” resistance. However, they use drug sensitive cancer cell line to do the experiments. This is not acceptable, as you need to prove the cell model you used is a MDR cell type with solid experimental data, such as the cytotoxic results between drug sensitive and drug resistance cell lines, or the over-expression of ABC transporters on the MDR cell line. For the experimental data reported in the present study, the author can only claim this formulation has cancer cell selective toxicity compared to the healthy cells, not relating to the “multidrug resistance”. Or the author can try to perform their study on a real MDR cancer cell line, such as A549/ADR, or MCF-7/ADR, and make a major revision.
Reviewer 3 Report
General Comments:
The manuscript submitted by Zlotnikov et al. describes selective sensitization of common cytotoxic by eugenol (EG) and apiol in A549 cells but diminished cytotoxicity in HEK293 cells. Moreover, the authors also showed increased intracellular concentration of doxorubicin in A549 cells but decreased cellular accumulation in HEK293 cells. .
While I appreciate the effort of the work presented, this manuscript has some problems as indicated below.
There are a lot of figures without replication numbers of experiments. I would like to point out that the authors have to apply several assays indicated in this manuscript to multiple cell lines, at least two cell lines each for cancer or non-cancer cell models.
Specific comments
1. In cancer pharmacology domain, doxorubicin, cisplatin and paclitaxel are classified as cytotoxic anti-cancer reagents rather than cytostatic.
2. Figure 3a and 3b, Figure 4, what are the unit for X-axis?
Figure 3c and 3d, for combination therapy studies, the cells should be treated with two compounds at a fixed-dose ratio and then interactions be assessed by calculating IC50 values and combination index (CI).
3. What are the rationales to study Dox 15-min accumulation in A549 (Figure 7) but 120-min accumulation in HEK293 cells (Table 2)?
4. Please indicate the statistic analysis methods and p-value for Figure 8.
Round 2
Reviewer 1 Report
Dear Authors,
I greatly appreciate your responses and your meticulous revisions.
Greetings,
Author Response
Dear Colleagues! The authors of the presented work sincerely thank you for the study of the article and writing a constructive review! All comments are taken into account.
Reviewer 2 Report
Problems are basically solved. Future advanced research in "real" MDR cancer cell is suggested.
Author Response

(The authors gave the same response as above.)
